# Neutralizing activity of BBIBP-CorV vaccine-elicited sera against Beta, Delta and other SARS-CoV-2 variants of concern

Xiaoqi Yu [1,8], Dong Wei [1,8], Wenxin Xu [1,8], Chuanmiao Liu [2,8], Wentian Guo[3], Xinxin Li[1], Wei Tan[1], Leshan Liu[3], Xinxin Zhang [1,3,9 ✉], Jieming Qu [4,5,6,9 ✉], Zhitao Yang [7,9 ✉] & Erzhen Chen [7,9 ✉]

The global pandemic of severe acute respiratory syndrome coronavirus 2 (SARS-CoV-2) has resulted in the generation of variants that may diminish host immune responses to vaccine formulations. Here we show a registered observational clinical trial (NCT04795414), we assess the safety and immunogenicity of the inactivated SARS-CoV-2 vaccine BBIBP-CorV in a cohort of 1006 vaccine recipients. No serious adverse events are observed during the term of the study. Detectable virus-specific antibody is measured and determined to be neutralizing in 698/760 (91.84%) vaccine recipients on day 28 post second vaccine dose and in 220/581 (37.87%) vaccine recipients on day 180 post second vaccine dose, whereas vaccine-elicited sera show varying degrees of reduction in neutralization against a range of key SARS-CoV-2 variants, including variant Alpha, Beta, Gamma, Iota, and Delta. Our work show diminished neutralization potency against multiple variants in vaccine-elicited sera, which indicates the potential need for additional boost vaccinations.

[1] Department of Infectious Diseases, Research Laboratory of Clinical Virology, Ruijin Hospital, Shanghai Jiao Tong University School of Medicine, Shanghai, China. [2] Department of Infectious Diseases, National Clinical Research Center for Infectious Diseases, Key Laboratory of Immunology in Chronic Diseases, First Affiliated Hospital of Bengbu Medical College, Anhui, China. [3] Clinical Research Center, Ruijin Hospital, Shanghai Jiao Tong University School of Medicine, Shanghai, China. [4] Department of Pulmonary and Critical Care Medicine, Ruijin Hospital, Shanghai Jiao Tong University School of Medicine, Shanghai, China. [5] Institute of Respiratory Diseases, Shanghai Jiao Tong University School of Medicine, Shanghai, China. [6] Shanghai Key Laboratory of Emergency Prevention, Diagnosis and Treatment of Respiratory Infectious Diseases, Shanghai, China. [7] Department of Emergency, Ruijin Hospital, Shanghai Jiao Tong University School of Medicine, Shanghai, China. [8] These authors contributed equally: Xiaoqi Yu, Dong Wei, Wenxin Xu, Chuanmiao Liu. [9] These authors jointly supervised this work: Xinxin Zhang, Jieming Qu, Zhitao Yang, Erzhen Chen. ✉email: zhangx@shsmu.edu.cn; jmqu0906@163.com; yangzhitao@hotmail.fr; chenerzhen@hotmail.com

G iven the unprecedented morbidity of the coronavirus disease 2019 (COVID-19), the efficacy of different vaccines needs to be assessed across diverse populations. The absence of immunity in the population causes susceptible people to be vulnerable to further waves of severe acute respiratory syndrome coronavirus 2 (SARS-CoV-2) infection, and healthcare workers are at a particularly high risk of infection. Sustained progress has been made in the development of SARS-CoV-2 vaccines, including inactivated vaccines[1–3], mRNA vaccines[4,5], adenovirus-vectored vaccines[6–8], and recombinant protein subunit vaccines[9], that are safe and exhibit immunogenicity against SARS-CoV-2. The inactivated vaccine BBIBP-CorV developed by Sinopharm, approved by the World Health Organization (WHO) for emergency use, is safe and well-tolerated in healthy people, and can induce high levels of neutralizing antibody titers to protect against SARS-CoV-2[1]. However, whether this vaccine could produce long-term protection is still under investigation.

The newly emerged SARS-CoV-2 variants of concern (VOC) and variants of interest (VOI) including Alpha (lineage B.1.1.7, first detected in the United Kingdom)[10], Beta (lineage B.1.351, first identified in South Africa)[11], Gamma (lineage P.1, initially expanded in Brazil)[12], and Iota (lineage B.1.526, largely found in South America)[13], are reportedly more efficiently and rapidly transported worldwide[14]. These variants contain mutations, such as N501Y and E484K in the receptor-binding domain (RBD) of spike glycoproteins, which are important for angiotensin-converting enzyme 2 (ACE2) binding and antibody recognition[15]. The highly transmissible Delta VOC (lineage B.1.617.2, first detected in India) recently emerged shows potential for immune escape[16] and the ability to evade vaccines[17]. Consequently, there is now great concern regarding the vaccine efficacy against these resistant variants.

Here, we report the safety and immunogenicity of an inactivated SARS-CoV-2 vaccine BBIBP-CorV and assess the 6-month durability of the humoral immune response in vaccine recipients, particularly evaluate the effect of multiple SARS-CoV-2 variants on vaccine-elicited neutralization. In brief, the BBIBP-CorV vaccine is safe and can effectively induce humoral responses in vaccine recipients. Neutralizing antibodies persist in 220/581 (37.87%) vaccine recipients 180 days after the second dose. Diminished neutralization potency against multiple variants is observed, indicating the potential need for additional boost vaccinations.

## Results

**Study participants**. Between January 14, 2021 and March 10, 2021, a total of 1006 healthcare workers in Shanghai Ruijin Hospital were recruited in this study. Figure 1 shows an overview of this study with the key time points and sample sizes at each time point. Among 1006 vaccine recipients, 284 were male and 722 were female, with a median age of 35.00 (28.00–43.00) years, and a total of 169 (16.80%) participants had at least one underlying disease. In addition, we included a panel of 571 naive individuals to ensure the accuracy of the specific antibody immunoassay and a panel of 16 COVID-19 recovered patients for the neutralization assay. The baseline characteristics of the study participants are shown in Table 1.

**Safety outcomes**. To date, no serious adverse events have been reported in this study. All adverse reactions were mild or moderate in severity and most cases were resolved by day seven after vaccination. A total of 447 (44.43%) of 1006 vaccine recipients experienced at least one adverse reaction after either dose. Common adverse reactions were reported more frequently after the second dose than after the first dose (Table 2). The overall

incidence of adverse reactions was 308 (30.62%) after the second dose and 241 (23.96%) after the first dose.

At least one local adverse reaction occurred after either dose in 258 (25.65%) of the 1006 vaccine recipients. The proportion of vaccine recipients who reported local adverse reactions increased after the second dose. Pain at the injection site was the most common local adverse reaction, which was reported by 231 (22.96%) participants, and was reported more frequently after the second dose (160 [15.90%]) than after the first dose (97 [9.64%]). Other adverse reactions at the injection site included redness (65 [6.46%]), swelling (50 [4.97%]), and rash (15 [1.49%]).

At least one systematic adverse reaction was reported by 310 (30.82%) of the 1006 vaccine recipients after either dose. The most common systematic adverse reaction was fatigue, which was reported by 206 (20.48%) vaccine recipients. Other systemic adverse reactions included headache (101 [10.04%]), diarrhea (34 [3.38%]), nausea and vomiting (31 [3.08%]), fever (27 [2.68%]), mucocutaneous abnormality (22 [2.19%]), myalgia and arthralgia (18 [1.79%]).

Clinical laboratory measurements revealed a few mild to moderate transient abnormalities. After the first dose, 39 (3.88%) vaccine recipients had decreased hemoglobin, 51 (5.07%) had an increased white blood cell count, two (0.20%) had an increased lymphocyte count, 14 (1.39%) had an increased neutrophils count, 31 (3.08%) showed increased alanine aminotransferase levels, 49 (4.87%) showed increased aspartate aminotransferase levels, 14 (1.39%) had increased serum total bilirubin levels, 40 (3.98%) had increased blood urea nitrogen levels, and two (0.20%) had increased creatinine levels. After the second dose, 27 (2.68%) vaccine recipients had decreased hemoglobin, 46 (4.57%) had an increased white blood cell count, four (0.40%) had an increased lymphocyte count, 15 (1.49%) had increased neutrophils count, 30 (2.98%) showed increased alanine aminotransferase levels, 35 (3.48%) showed increased aspartate aminotransferase levels, two (0.20%) had increased serum total bilirubin levels, 27 (2.68%) had increased blood urea nitrogen levels, and four (0.40%) had increased creatinine levels. None of the post-vaccination abnormalities were considered clinically significant.

**Immunogenicity responses**. Immunological analyses were performed among individuals from whom blood samples were collected at each time point. The number, sex and age of participants at each time point are shown in Supplementary Table 1. Specific antibodies against SARS-CoV-2 were also assessed in a panel of 571 naive individuals. Among them, four had relatively low antibody titers (1.12, 1.52, 1.94, and 2.26, respectively).

Rapid antibody responses to SARS-CoV-2 were observed in 609 (63.17%) of 964 individuals from whom blood samples were collected on day 21 after the first dose (V3), the median antibody level was 5.32 (2.33–13.35) (Fig. 2a). Enhanced specific antibody responses against SARS-CoV-2 were detected in 731 (96.18%) of 760 vaccine recipients who had blood samples taken on day 28 after the second dose (V4), with median antibody level at 33.96 (12.56–82.04) (Fig. 2a), which was an increase relative to the antibody responses on day 21 after the first dose. The seroconversion rate of neutralizing antibodies against the wild-type strain was 698 (91.84%) of 760 individuals, and the geometric mean titer (GMT) was 62.68 (95% confidence interval [CI] 57.02–68.91) (Fig. 2b). Sex was not a factor affecting the induction of neutralizing antibodies after the second dose. Vaccine recipients with seroconversion of neutralizing antibodies on day 28 after the second dose were significantly younger than those without seroconversion (median age: 36.00 (30.00–43.00) vs. 41.50 (32.75–49.00), $p = 0.003$).

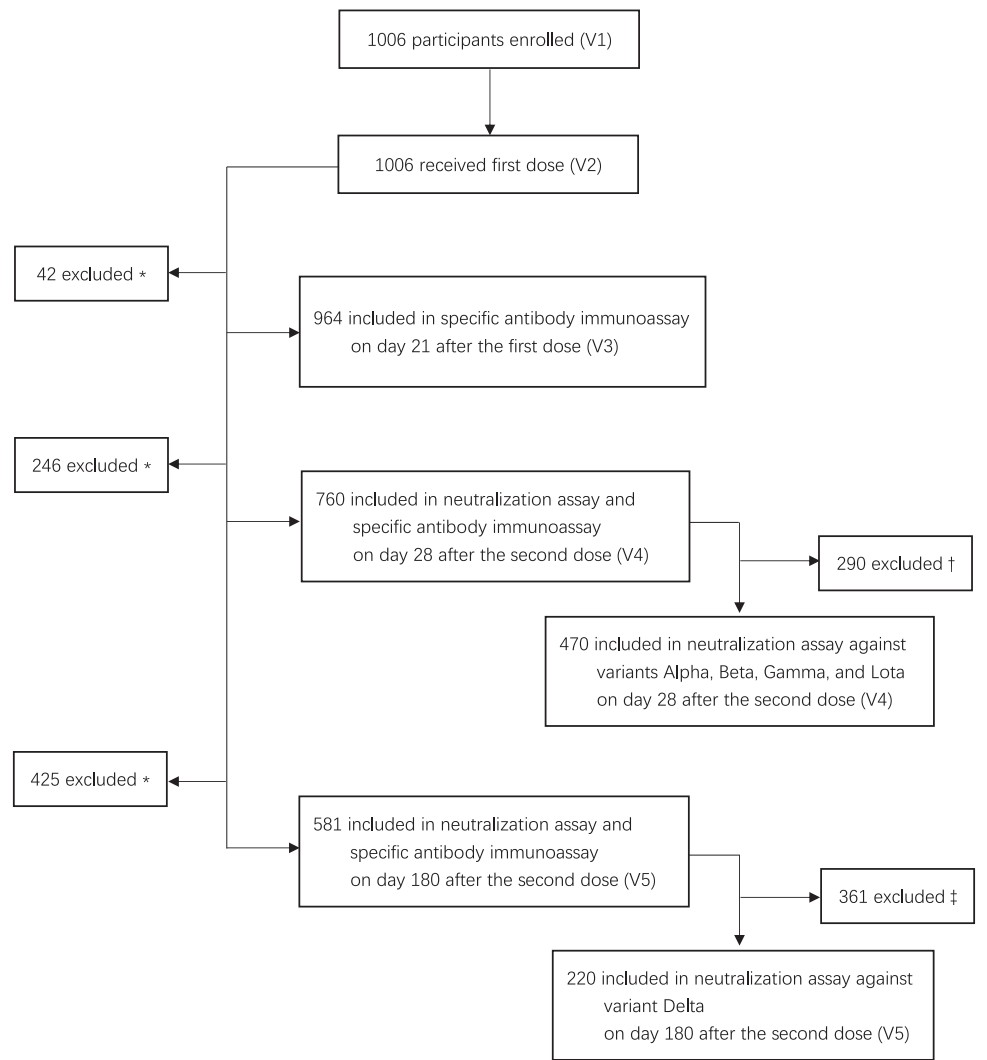

**Fig. 1 Study profile.** *Participants who were administered the vaccination and completed all safety visits, but did not have blood samples taken upon personal request. †290 participants who showed negative neutralizing activity against the wild-type strain or who refused to undergo testing the neutralization assay against multiple variants on day 28 after the second dose were excluded. ‡361 participants with negative neutralizing activity against the wild-type strain on day 180 after the second dose were excluded.

**Table 1 Baseline demographic characteristics of all participants.**

|  | Vaccine recipients ($n = 1006$) | Naive individuals ($n = 571$) | Convalescent patients ($n = 16$) |
|---|---|---|---|
| Age | 35.00 (28.00–43.00) | 34.00 (27.00–43.00) | 49.00 (41.00–55.00) |
| Sex |  |  |  |
| Male | 284 (28.23%) | 161 (28.20%) | 11 (68.75%) |
| Female | 722 (71.77%) | 410 (71.80%) | 5 (31.25%) |
| Underlying diseases | 169 (16.80%) | 74 (12.96%) | 5 (31.25%) |
| Diabetes | 12 (1.19%) | 7 (1.23%) | 4 (25.00%) |
| Hypertension | 74 (7.36%) | 29 (5.08%) | 2 (12.50%) |

Data are expressed as median (interquartile range [IQR]) or number (%).

Among the 760 vaccine recipients, 470 participants with positive neutralizing activity against the wild-type strain (GMT 68.72, [95% CI 61.97–76.20]) agreed to undergo testing for neutralizing activity against multiple variants. Neutralization assays were performed against four circulating SARS-CoV-2 variants in 470 vaccine-elicited sera on day 28 after the second dose (V4). 57 (12.13%) of the 470 vaccine-elicited sera showed a complete loss of neutralizing activity against variant Alpha. The

neutralization GMT against the Alpha variant decreased by 2.2-fold to 31.17 (95% CI 27.71–35.07) compared with that of the wild-type strain (Fig. 3a). In addition, 114 (24.26%) and 99 (21.06%) of the 470 vaccine-elicited sera showed a complete loss of neutralizing activity against the Gamma and Iota variants, respectively. The GMT against the Gamma variant decreased by 1.9-fold to 37.07 (95% CI 32.84–41.84), whereas a marked decrease by 3.8-fold was observed in the GMT against the Iota

**Table 2 Adverse reactions after the first and second doses of vaccination.**

|  | Either dose | Dose 1 | Dose 2 |
|---|---|---|---|
| Total adverse reactions | 447 (44.43%) | 241 (23.96%) | 308 (30.62%) |
| Any local symptoms | 258 (25.65%) | 117 (11.63%) | 175 (17.40%) |
| Pain | 231 (22.96%) | 97 (9.64%) | 160 (15.90%) |
| Redness | 65 (6.46%) | 56 (5.57%) | 11 (1.09%) |
| Swelling | 50 (4.97%) | 24 (2.39%) | 28 (2.78%) |
| Rash | 15 (1.49%) | 12 (1.19%) | 5 (0.50%) |
| Any systemic symptoms | 310 (30.82%) | 177 (17.59%) | 188 (18.69%) |
| Fatigue | 206 (20.48%) | 88 (8.75%) | 134 (13.32%) |
| Headache | 101 (10.04%) | 72 (7.16%) | 45 (4.47%) |
| Fever | 27 (2.68%) | 19 (1.89%) | 11 (1.09%) |
| Diarrhea | 34 (3.38%) | 21 (2.09%) | 16 (1.59%) |
| Nausea and vomiting | 31 (3.08%) | 28 (2.78%) | 4 (0.40%) |
| Mucocutaneous abnormality | 22 (2.19%) | 15 (1.49%) | 10 (0.99%) |
| Myalgia and arthralgia | 18 (1.79%) | 14 (1.39%) | 5 (0.50%) |

Data are expressed as number (%).

variant (18.12 [95% CI 16.33–20.10]). Neutralizing activity against the Beta variant was preserved in only 199 (42.34%) of the 470 vaccine-elicited sera, with a significantly reduced GMT (15.08, [95% CI 13.06–17.42]) compared with that of the wild-type strain.

We further analyzed the cross-reactivity of neutralizing antibodies against the four variants. As shown in Fig. 4, on day 28 after the second dose, a total of 163 (34.68%) of the 470 vaccine-elicited sera preserved neutralizing activity against all four variants including Alpha, Beta, Gamma, and Iota. Only 14 (2.98%) of the 470 vaccine-elicited sera did not induce neutralizing antibodies against any of these four variants. Among 199 (42.34%) of the 470 vaccine-elicited sera with positive neutralizing antibodies against the Beta variant, only one had negative neutralizing activity against the Alpha, Gamma, and Iota variants.

We next assessed the durability of the humoral immune response in 581 participants who were followed up on day 180 after the second dose (V5). Specific antibodies against SARS-CoV-2 could still be detected in 500 (86.06%) of the 581 vaccine recipients, although the median antibody level decreased to 6.08 (2.87–14.86) (Fig. 2a). However, significantly fewer participants had quantifiable neutralizing antibodies on day 180 after the second dose, comprising 220 (37.87%) of the 581 individuals, and the GMT decreased to 40.84 (95% CI 33.73–49.45) (Fig. 2b). The neutralization assay was also performed against the currently most prevalent variant Delta in 220 participants who showed positive neutralizing activity against the wild-type strain. Neutralizing activity against the Delta variant remained detected in 96 (43.64%) of 220 participants, and the GMT showed a 2.6-fold reduction (15.93 [95% CI 12.72–19.94]) compared with that of the wild-type strain (Fig. 3b).

In comparison, a panel of 16 convalescent sera collected at sixth-month post symptom onset from COVID-19 recovered patients revealed a long-lasting humoral immune response, with a median SARS-CoV-2 specific antibody level of 246.80 (89.09–366.14) (Fig. 2a). Moreover, neutralizing antibody responses were detected in all convalescent patients, and the GMT was 404.06 (95% CI 250.74–651.12), which was significantly higher than that in the vaccine recipients on day 28 after the second dose (Fig. 2b). All 16 (100%) convalescent sera were able to neutralize the Alpha, Delta, and Iota variants, with varying degrees of reduction in neutralization. Compared with the wild-type strain, variant Alpha, Iota, and Delta were 1.9-fold, 2.3-fold, and 3.4-fold less sensitive to sera, respectively (Fig. 3c). We found that two (12.50%) and one (6.25%) of the 16 convalescent sera had completely lost neutralizing activity against the Beta and

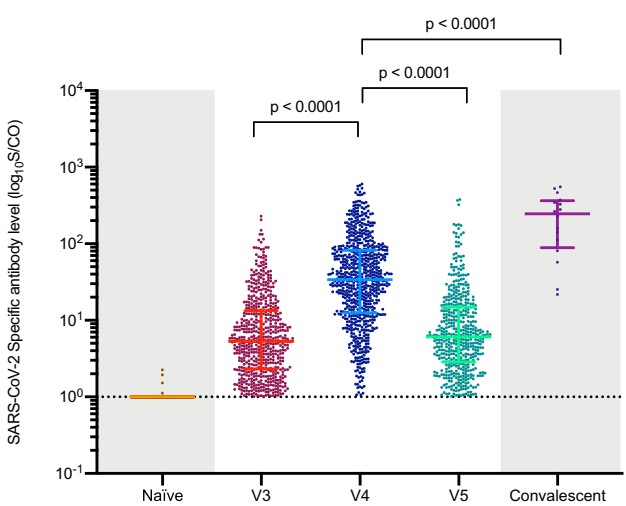

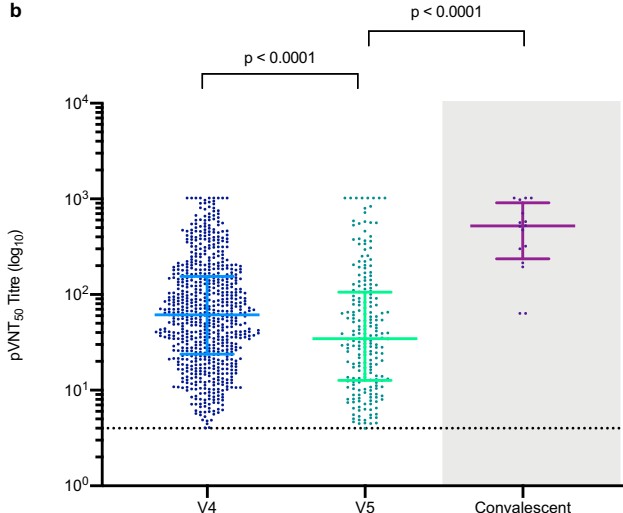

**Fig. 2 SARS-CoV-2 specific antibody and neutralizing antibody responses. a** Specific antibodies against SARS-CoV-2 in 964 vaccine recipients on day 21 after the first dose (V3), 760 vaccine recipients on day 28 after the second dose (V4), and 581 vaccine recipients on day 180 after the second dose (V5). The shaded portions indicate two categories of reference values: SARS-CoV-2 specific antibody levels of 571 naive individuals (Naive) and 16 convalescent sera collected at sixth-month post symptom onset from COVID-19 recovered patients (Convalescent). The horizontal dashed line represents the lower limit of detection for the assay (>1). Error bars indicate median and interquartile range (IQR). **b** The results of 50% pseudovirus neutralization titer (pVNT50) against the wild-type strain in 760 vaccine recipients on day 28 after the second dose (V4), and 581 vaccine recipients on day 180 after the second dose (V5). The shaded portion indicates the category of control values: 50% pseudovirus neutralization titer (pVNT50) against the wild-type strain from 16 convalescent COVID-19 patients. The horizontal dashed line represents the lower limit of detection for the assay (>4). Error bars represent the geometric mean with the 95% confidence interval (95% CI). The exact p values (two-sided) were calculated using the Mann–Whitney U test. No adjustment was done for multiple comparison. Source data are provided as a Source data file.

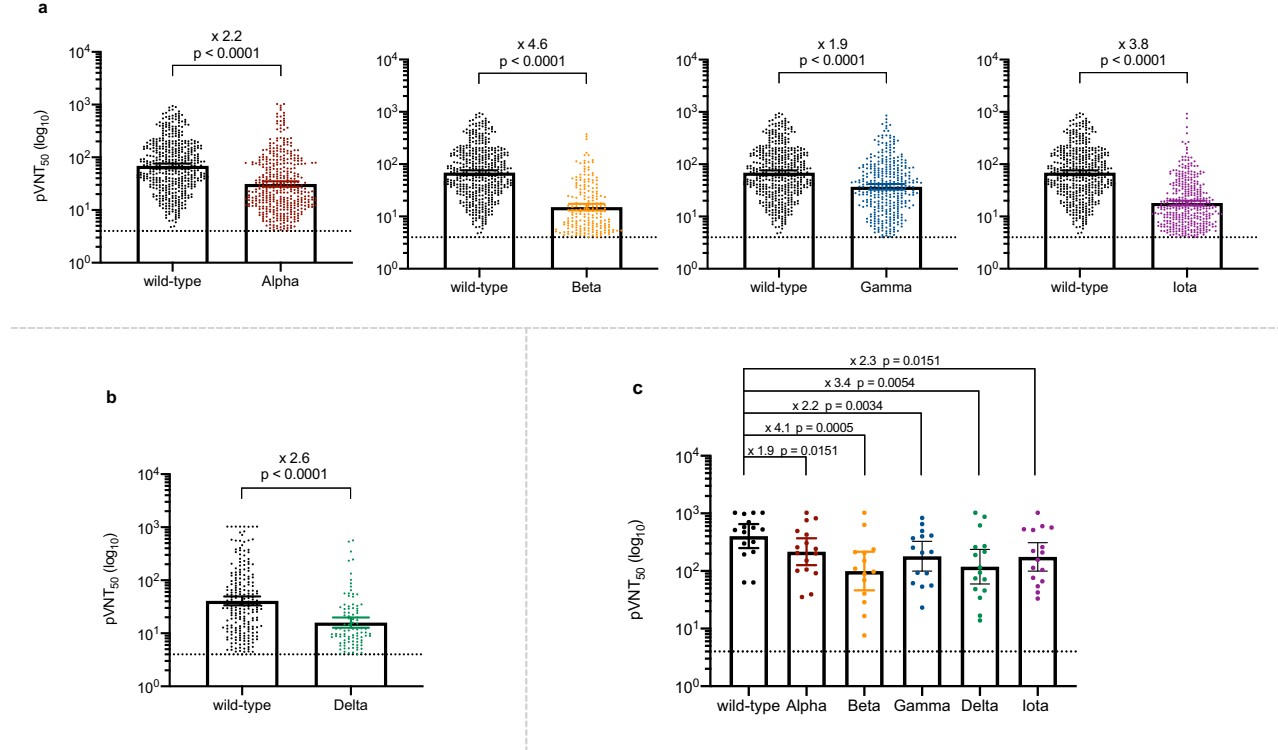

**Fig. 3 Neutralizing antibody responses against multiple SARS-CoV-2 variants. a** Results of 50% pseudovirus neutralization titer (pVNT50) in 470 vaccine recipients on day 28 after the second dose against multiple SARS-CoV-2 variants, including Alpha ($p < 0.0001$), Beta ($p < 0.0001$), Gamma ($p < 0.0001$), and Iota ($p < 0.0001$) variants compared with the wild-type strain. **b** Results of 50% pseudovirus neutralization titer (pVNT50) in 220 vaccine recipients on day 180 after the second dose against Delta ($p < 0.0001$) compared the wild-type strain. **c** Results of 50% pseudovirus neutralization titer (pVNT50) in 16 convalescent COVID-19 patients against multiple SARS-CoV-2 variants, including Alpha ($p = 0.0151$), Beta ($p = 0.0005$), Gamma ($p = 0.0034$), Delta ($p = 0.0054$), and Iota ($p = 0.0151$) variants compared with the wild-type strain. Each vertical bar represents the geometric mean with the 95% confidence interval (95% CI). The fold-changes in geometric mean titer are shown above. The horizontal dashed line represents the lower limit of detection of the assay (>4). The exact $p$ values (two-sided) were calculated using the Wilcoxon matched-pairs signed-rank test. No adjustment was done for multiple comparison. Source data are provided as a Source data file.

Gamma variants, respectively. Compared with the wild-type strain, the GMT decreased by 4.1-fold and by 2.2-fold against the Beta and Gamma variants, respectively, in convalescent sera.

**Cytokine responses.** Dynamic changes in several key inflammatory cytokines, including interferon-γ (IFN-γ), interleukin (IL)-10, IL-12p70, IL-13, IL-2, IL-6, IL-8, and tumor necrosis factor-α (TNF-α), were tested in serum at different time points. The levels of some cytokines showed notable changes from the first dose through 28 days after the second dose (Fig. 5). On day 21 after the first dose and on day 28 after the second dose, the levels of IFN-γ, IL-10, and IL-13 were significantly higher than their levels on the day of the first dose. The levels of IL-8 and TNF-α showed significant increases on day 21 after the first dose, followed by significant decreases on day 28 after the second dose.

The cytokine responses were lower among participants who did not successfully induce neutralizing antibodies against SARS-CoV-2, including the levels of IFN-γ and TNF-α on day 28 after the second dose, and the level of IL-12p70 on the day of the first dose (Fig. 6).

## Discussion

In this study, the inactivated vaccine BBIBP-CorV was safe and well-tolerated in participants. No serious adverse events have been reported to date, and all the adverse reactions were mild or moderate. The most frequently reported local adverse reaction was pain at the injection site, and the most common systematic

adverse reaction was fatigue. Clinical laboratory measurements revealed a few mild to moderate transient abnormalities, but none of the post-vaccination abnormalities were considered clinically significant.

The specific antibody assay against SARS-CoV-2 was performed to assess the humoral immune responses. To ensure the accuracy of this assay, we included a panel of 571 naive individuals with neither prior COVID-19 symptoms nor a history of SARS-CoV-2 vaccination. Among them, four had relatively low antibody titers, which may be due to the false-positive results caused by non-specific endogenous or exogenous interferents. However, the possibility of subclinical infection cannot be ruled out, despite the effective intervention and control of the COVID-19 pandemic in Shanghai, China.

The neutralization assay using lentivirus-based SARS-CoV-2 pseudoviruses was performed to assess the resistance of multiple SARS-CoV-2 variants. The traditional neutralization assay for the SARS-CoV-2 vaccines using the isolated live virus must be performed at biosafety level 3 facilities, whereas the pseudotype virus-based neutralization assay against SARS-CoV-2 has been developed and can be handled in biosafety level 2 facilities[18]. A previous study revealed a high degree of concordance between the pseudotype neutralization assay and isolated live SARS-CoV-2 neutralization assay[19], suggesting that the pseudotype neutralization assay can be used to evaluate the inhibitory effect of the vaccines on viral attachment and entry[20]. Therefore, we used a safer pseudovirus-based neutralization assay to evaluate neutralizing activity in this study.

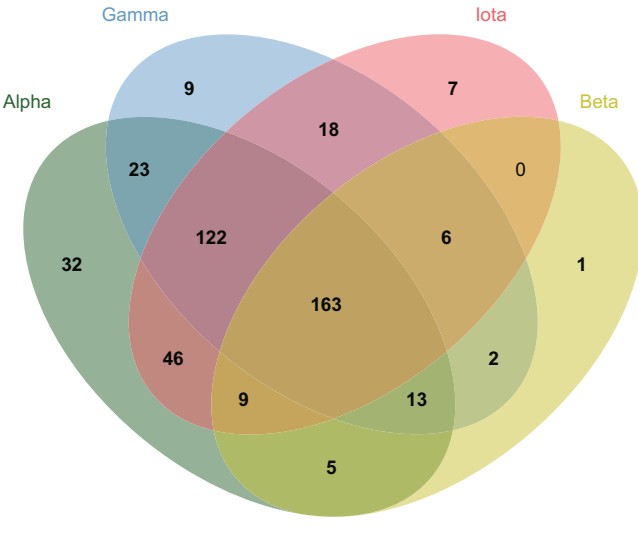

**Fig. 4 Cross-reactivity of neutralizing antibodies against four variants.**
Different colors indicate four different SARS-CoV-2 variants: green, yellow, blue, and red indicate the Alpha, Beta, Gamma, and Iota variants, respectively. Numbers on the diagram indicate the number of participants with positive neutralizing activity against each variant.

Neutralizing antibody responses against SARS-CoV-2 were successfully induced in 698 (91.84%) of the 760 individuals on day 28 after the second dose, which is lower than the previously reported seroconversion rate[1]. Although previous data indicated that factors such as age, sex, and the presence of a coexisting condition does not affect the efficacy of specific COVID-19 vaccine[4], in this study, younger age was significantly related to the seroconversion of neutralizing antibodies. Healthcare workers tend to sleep less and have more irregular circadian rhythm than other populations, whether sleep pattern impact the vaccine-elicited antibody response remains unclear. It was reported that sleep may boost virus-specific adaptive immunity and promote a cytokine milieu that supports the cellular response, additionally, lack of sleep during the night after vaccination was found to reduce the antibody response to hepatitis A, hepatitis B, and influenza vaccinations[21]. Thus, extending sleep duration at night after vaccination may result in a higher antibody response, and further studies are needed to provide more conclusive evidence on the production of neutralizing activity.

The emergence of new SARS-CoV-2 variants has led to concerns regarding the potential of these variants to circumvent immunity elicited by natural infection or vaccination. In this study, we tested neutralization of BBIBP-CorV vaccine-elicited sera against a range of key SARS-CoV-2 variants, including variant Alpha, Beta, Gamma, Iota, and Delta. The Alpha variant, a highly transmissible variant of concern, consists of a series of mutations including N501Y, which is located in the receptor-binding domain of the spike protein[10]. Variants containing this mutation bind more tightly to the cellular receptor, angiotensin-converting enzyme 2 (ACE2)[22]. The Beta variant of concern carries the immune escape-associated mutation E484K[23]. Previous research showed that many highly neutralizing monoclonal antibodies, most convalescent sera, and mRNA-induced immune

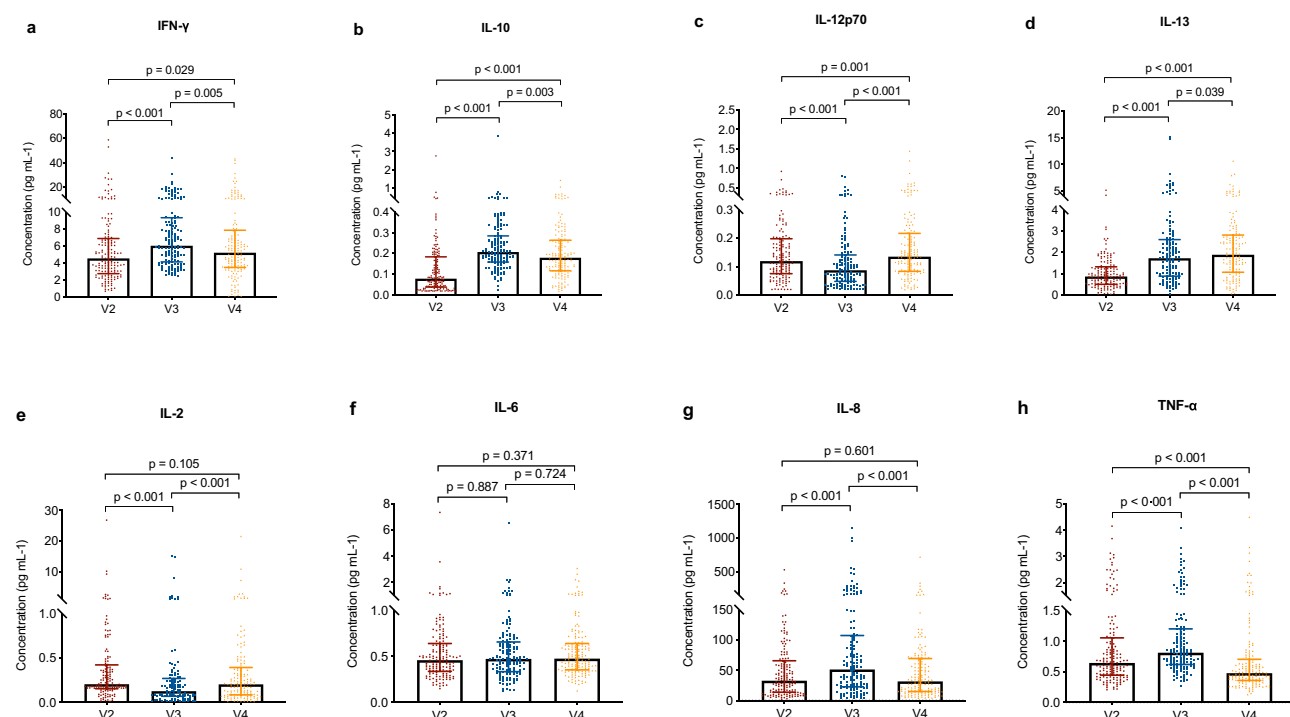

**Fig. 5 Dynamic changes in the levels of key inflammatory cytokines. a–h** Levels of key inflammatory cytokines including interferon-γ (IFN-γ), interleukin (IL)-10, IL-12p70, IL-13, IL-2, IL-6, IL-8, and tumor necrosis factor-α (TNF-α) were measured in 154 vaccine recipients on the day of the first dose (V2), on day 21 after the first dose (V3), and on day 28 after the second dose (V4). Box plots indicate the median and interquartile range (IQR). The exact $p$ values (two-sided) were calculated using the Wilcoxon signed-rank test. IL: interleukin. No adjustment was done for multiple comparison. Source data are provided as a Source data file.

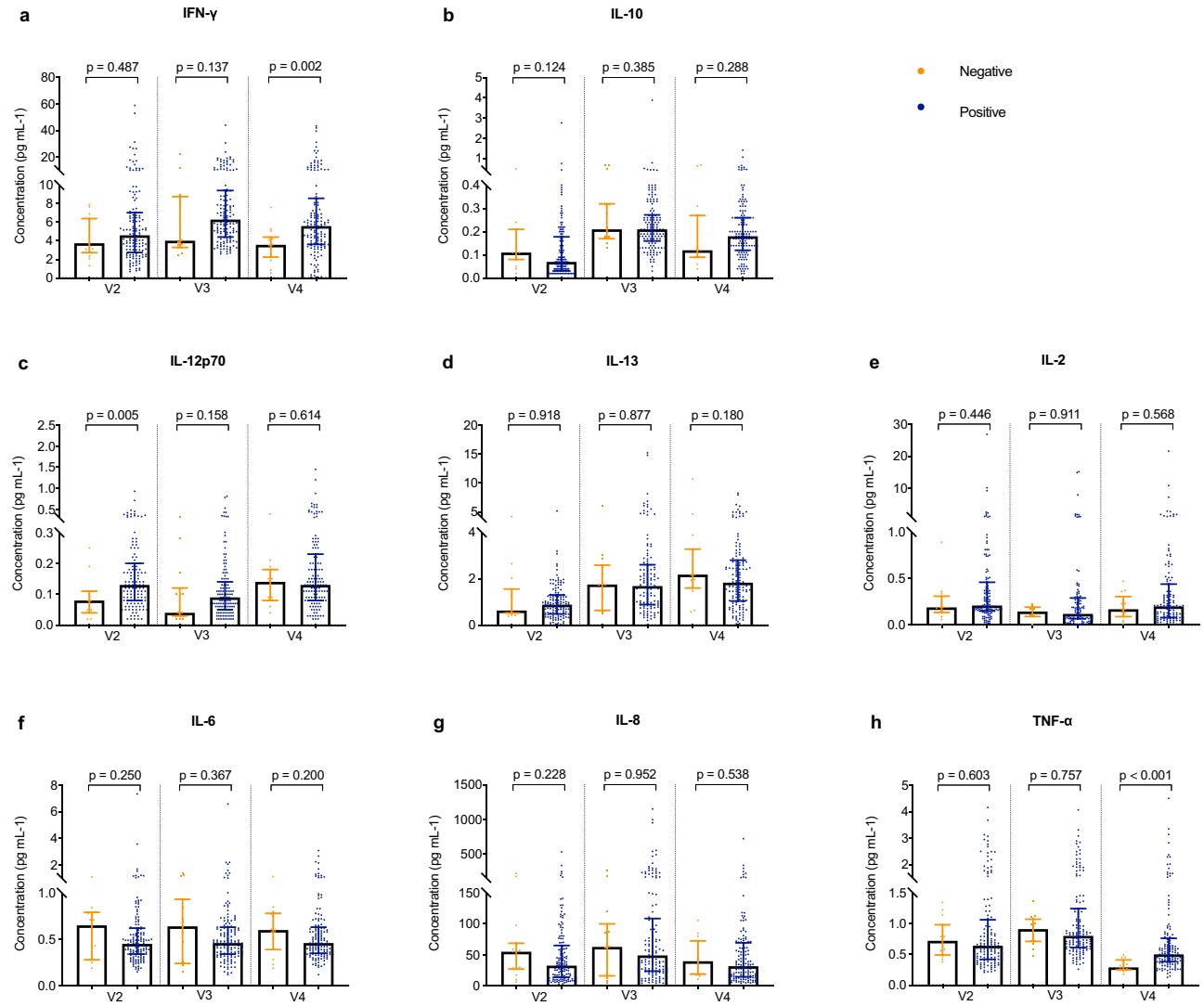

**Fig. 6 Levels of key inflammatory cytokines in relation to neutralizing antibody responses. a–h** Levels of key inflammatory cytokines including interferon-γ (IFN-γ), interleukin (IL)-10, IL-12p70, IL-13, IL-2, IL-6, IL-8, and tumor necrosis factor-α (TNF-α) on the day of the first dose (V2), on day 21 after the first dose (V3), and on day 28 after the second dose (V4) according to neutralizing antibody responses. Blue dots indicate 139 vaccine recipients with positive neutralizing activity on day 28 after the second dose. Yellow dots indicate 15 vaccine recipients with negative neutralizing activity on day 28 after the second dose (V4). Box plots indicate the median and interquartile range (IQR). The exact p values (two-sided) were calculated using the Mann–Whitney U test. IL: interleukin. No adjustment was done for multiple comparison. Source data are provided as a Source data file.

sera exhibited reduced inhibitory activity against viruses containing the E484K mutation[24]. The Gamma variant of concern[12] and Iota variant of interest[13] are both highly transmissible variants containing the E484K mutation. The newly identified variant of concern Delta contains diverse spike mutations, including mutation L452R[25], which has shown resistance to some monoclonal antibodies and sera[26,27].

Recent reports have indicated that mRNA vaccine-elicited sera largely preserved neutralizing titers against the Alpha variant[28,29]. However, significant reductions in the titers of neutralizing antibodies against the Beta variant were observed in mRNA vaccine-elicited sera[30,31]. Our study provides data on the neutralizing activity against the Alpha, Beta, Gamma, and Iota variants on day 28 after the second dose in 470 BBIBP-CorV-elicited sera. In a substantial proportion of vaccine-elicited sera, a complete loss of neutralizing activity against the Alpha variant with a decrease in the GMT was observed, consistent with a previous study demonstrating a marked decrease in the GMT in neutralization of the Alpha variant[32]. In multiple studies, the Beta

variant showed the strongest resistance to neutralization in convalescent or vaccinee sera[33]. According to a previous report that included 25 participants administered the BBIBP-CorV vaccine, 20 (80%) serum samples showed complete or partial loss of neutralization against the Beta variant[32]. Similarly, we found that the vaccine-elicited sera were significantly less effective in neutralizing the Beta variant. In addition, 114 (24.26%) and 99 (21.06%) vaccine-elicited sera showed a complete loss of neutralizing activity against the Gamma and Iota variants, respectively. As expected, more participants showed positive neutralizing activity against the Alpha variant than against the other three variants carrying the immune escape-associated mutation E484K, however, why the Beta variant showed stronger resistance than the Gamma and Iota variants remains unclear. Previous studies also showed that the Beta variant is more refractory to neutralization than the Gamma variant[31,34]. The possible explanation is that N-terminal domain (NTD) substitutions in the Beta variant contribute to neutralization escape[35], whereas other undefined mutations in the Gamma and Iota

variants may compensate for the negative effects on inhibition of E484K/N501Y mutations based on the previous research[24].

We also tested the neutralizing activity against the currently most prevalent variant Delta on day 180 after the second dose in vaccine-elicited sera. It has been reported that sera from recovered patients and inactivated vaccine recipients showed significantly reduced neutralization against the Delta variant[36], and lower neutralizing titers were observed against the Delta variant relative to the wild-type strain in vaccinated individuals[25,34,37]. In this study, significantly fewer participants had quantifiable neutralizing antibodies against the Delta variant, which is similar to previous data[38].

Most previous studies focused on the neutralizing activity of vaccinee sera soon after vaccination, thus, the durability of protection induced by the inactivated vaccine is currently unknown. mRNA vaccine-induced antibody activity remained high in 33 healthy adult participants 180 days after the second dose[39]. In our study, neutralizing antibodies elicited by the vaccine persisted for 6 months after the second dose in 220 (37.87%) of the 581 vaccine recipients, suggesting that a booster dose is needed to extend the duration of neutralizing activity against emerging variants. However, it should be noted that neutralization is a part of the humoral immune response, which does not account for all potentially protective vaccine responses. A recent report showed that the mRNA vaccine elicited a strong CD4 cytokine response involving type 1 helper T (Th1) cells[40,41] and a robust CD8 T cell response[42], and natural SARS-CoV-2 infection may induce a memory B-cell response[43], therefore, inactivated vaccination may induce efficient memory cellular responses despite waning neutralizing activity.

This study also has several limitations. First, our study relies on lentivirus-based SARS-CoV-2 pseudoviruses, which can only model viral entry. Moreover, the contribution of additional mutations other than the spike protein to neutralization resistance in these variants cannot be confirmed. Second, our study population did not cover individuals from more susceptible groups in all ages. Caution should be taken when extrapolating our findings to adolescent, older adults, or people with pre-existing diseases. Third, the cellular immunity was not comprehensively assessed in this study. Further work should provide greater insight into the role of cell-mediated immunity in vaccine efficacy against the emerging variants. Fourth, although the inactivated vaccine BBIBP-CorV elicited neutralizing antibody responses in the majority of participants, the real-world efficacy of the vaccine against the emerging variants remains to be determined. Fifth, new SARS-CoV-2 variants continuously emerge, the variants of current concern constantly change. Nevertheless, neutralization against a range of key variants of particular interest was assessed in this study.

In conclusion, the data presented here contribute to the evidence of neutralization of the inactivated vaccine against known predominant SARS-CoV-2 variants. Overall, the inactivated SARS-CoV-2 vaccine BBIBP-CorV was safe and well-tolerated in the recruited healthcare workers, and rapid humoral responses were induced after the first dose vaccination. A total of 698 (91.84%) of the 760 participants had neutralizing antibodies against SARS-CoV-2 on day 28 after the second dose, and 220 (37.87%) showed preservation of the neutralizing activity on day 180 after the second dose. Diminished neutralization potency against multiple variants was also observed, suggesting the potential need for additional boost vaccinations.

## Methods

**Study design and participants**. This study is registered with ClinicalTrials.gov, NCT04795414. The study protocol is available in the supplementary information file. The study protocol and informed consent were approved by the Ethics

Committee of Shanghai Ruijin Hospital (RJHKY2021-12) in accordance with the Declaration of Helsinki and Good Clinical Practice. Written informed consent was obtained from all participants before the screening. From January 14, 2021 to March 10, 2021, healthcare workers in Shanghai Ruijin Hospital, aged 18–59 years, with negative serum specific antibodies against SARS-CoV-2 at the time of screening (V1), and willing to receive two doses, 21 days apart of inactivated SARS-CoV-2 vaccine (BBIBP-CorV, Sinopharm) were eligible participants and were recruited in this study. Blood samples were collected from the participants for serology tests on the day of the first dose (V2), on day 21 after the first dose (V3), and on day 28 after the second dose (V4), and on day 180 after the second dose (V5). Naive individuals were defined as those who had neither prior COVID-19 symptoms nor a history of SARS-CoV-2 vaccination. They were 18–59 years of age, and tested negative for SARS-CoV-2 viral nucleic acid. Study participants did not receive any compensation.

**Convalescent serum panel**. A panel of 16 convalescent sera collected at sixth-month post symptom onset was selected by matching age from a cohort of 22 COVID-19 recovered patients in a published study[44]. All individuals were 18–59 years of age, had previously tested positive for the SARS-CoV-2 viral nucleic acid, were diagnosed with COVID-19, without a history of SARS-CoV-2 vaccination at the time of sampling. This panel included six moderate cases, eight severe cases, and two critical cases. The sera were kindly provided by the First Affiliated Hospital of Bengbu Medical College (Anhui, China).

**Safety assessments**. Each participant was informed of potential vaccine side effects and encouraged to report them. All relevant clinical data were collected from the hospital electronic system. Solicited local and systemic reactions were reported by the participants and recorded in an electronic diary, unsolicited adverse events and serious adverse events were assessed starting after administration of each dose. Laboratory safety measurements including hemoglobin, white blood cell count, lymphocyte count, neutrophils count, platelets, alanine aminotransferase, aspartate aminotransferase, serum total bilirubin, serum albumin, creatinine, and blood urea nitrogen levels were tested within 28 days post each dose.

**Immunogenicity assessments**. Specific antibodies against SARS-CoV-2 were measured using a chemiluminescence kit manufactured by Wantai BioPharm (China). The antibody levels were expressed as the chemiluminescence signal according to the manufacturer's instructions.

**Pseudovirus-based neutralization assay**. A pseudovirus-based neutralization assay was performed as previously described[18] with minor modifications. The lentivirus-based SARS-CoV-2 pseudoviruses expressing a luciferase reporter gene and bearing the spike protein from the wild-type strain (Wuhan-Hu-1), variant Alpha (lineage B.1.1.7), variant Beta (lineage B.1.351), variant Gamma (Lineage P.1), variant Iota (Lineage B.1.526), and variant Delta (Lineage B.1.617.2) tested in this study were manufactured by Vazyme Biotech Co., Ltd. (China). In brief, to assess the neutralization geometric mean titers (GMTs) of vaccine-elicited sera, six serial dilutions of heat-inactivated sera (in a four-fold step-wise manner) were incubated with 250 $TCID_{50}$ SARS-CoV-2 pseudoviruses per well for 1 h, together with the virus control and cell control wells, before seeding 20,000 HEK293T-ACE2 cells per well in 96-well plates. After 48 h of incubation in a 5% $CO_2$ environment at 37 °C, the supernatant was removed, and the luminescence was measured using Luciferase Assay System (Promega Biotech Co., Ltd) according to the manufacturer's instructions. The 50% inhibitory concentration ($IC_{50}$) was defined as the serum dilution at which the relative light units (RLUs) were reduced by 50% compared with the virus control wells after subtraction of background RLUs in the cell control wells with only cells. The $IC_{50}$ values were calculated by generating a three-parameter non-linear regression curve fit in GraphPad Prism 8.4.0. A neutralizing antibody potency of <1:4 was considered as a negative result.

**Cytokine measurement**. Inflammatory cytokines including interferon-γ (IFN-γ), interleukin-10 (IL-10), IL-12p70, IL-13, IL-2, IL-6, IL-8, and tumor necrosis factor-α (TNF-α) were measured using an electro-chemiluminescent multiplex assay (Meso Scale Discovery) to explore the underlying immune responses.

**Outcomes**. The primary safety endpoint was any adverse reaction within 28 days after each dose of vaccination. The secondary safety endpoint was any clinical laboratory abnormality within 28 days after each dose of vaccination. The primary immunogenic endpoints were the seroconversion rate and the titers of specific antibodies and neutralizing antibodies against SARS-CoV-2 on day 28 post the second dose. The secondary immunogenic endpoints were the seroconversion rate and the titers of specific antibodies and neutralizing antibodies against SARS-CoV-2 on day 180 post the second dose.

**Statistical analysis**. Continuous variables that were not normally distributed were presented as median (interquartile range [IQR]). Categorical variables were described as count (%). The statistical method used for analysis of antibody titers is

the geometric mean (GMTs) and the corresponding 95% confidence interval (95% CI). The values were compared by the Wilcoxon signed-rank test and the Mann–Whitney U test as appropriate. The chi-square test ($\chi^2$) was applied to assess the distribution in different groups. Graphs were plotted using GraphPad Prism 8.4.0 software. Venn diagrams were drawn using jvenn: an interactive Venn diagram viewer[45]. Statistical analyses were performed using SPSS 24.0 software. A two-sided $p$ value of <0.05 was considered statistically significant.

**Reporting summary**. Further information on research design is available in the Nature Research Reporting Summary linked to this article.

## Data availability

The authors declare that all data supporting the findings of this study are available within the article and its Supplementary Information files. Source data are provided with this paper.

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

## Acknowledgements

This study was funded by Shanghai Key Laboratory of Emergency Prevention, Diagnosis and Treatment of Respiratory Infectious Diseases (20dz2261100) awarded to J.Q., and a grant from Science and Technology Commission Shanghai Municipality (No. 20JC1410200) awarded to X.Z. The investigators express their gratitude to Prof. Guang Ning for his great support. We thank all the participants involved in this study.

## Author contributions

X.Z., J.Q., Z.Y., and E.C. had the idea for and designed the study. X.Y., C.L., W.X., W.G., and L.L. were responsible for collecting and summarizing the clinical data. D.W., W.X., X.Y., X.L., and W.T. performed the experimental studies. X.Y., D.W., W.X., and W.G. carried out the analysis. X.Y. and X.Z. drafted the manuscript. All authors reviewed and approved the final version.

## Competing interests

The authors declare no competing interests.
