## [Peer Review File · Nature Communications]

Neutralizing activity of BBIBP-CorV vaccine-elicited sera against Beta, Delta and other SARS-CoV-2 variants of concernREVIEWER COMMENTS

Reviewer #1 (Remarks to the Author):

This manuscript attempts to characterize cross-reactive responses to SARS-CoV-2 emerging variants after vaccination with an inactivated virus vaccine.

1. The manuscript is severely under referenced and several sentences contain claims without any citations. The authors should consider citing primary literature more.
2. In figure 1, it becomes clear that antibody responses mature and increase after 2nd dose. However, the authors should include negative controls (naive individuals), convalescent individuals (previously infected) as well as individuals who received a different vaccine. This will really be the ultimate test of the assay setup as well as tell us how well the immune responses compare to other vaccines.
3. It is very surprising that the neutralization activity was significantly decreased against the B.1.1.7 pseudovirus. This is contrary to several other reports which use samples from mRNA vaccine recipients. To strengthen the claim, the authors should include 10 or more convalescent individuals in these experiments.
4. How do the authors feel about the drop of neutralization against B.1.351 but not P.1? They both are very similar in terms of mutations. Again, positive controls are necessary to strengthen these claims.

Reviewer #2 (Remarks to the Author):

In this manuscript, Yu et al. assessed the safety and immunogenicity of an inactivated SARS-CoV-2 vaccine 48 BBIBP-CorV, and measured the vaccine's efficacy against four global variants of concern: Lineage B.1.1.7, Lineage B.1.351, Lineage P.1, and Lineage B.1.526 to determine the neutralizing activity of vaccine-elicited sera. This is a very important study since the BBIBP-CorV vaccine has been approved and is being used for vaccinations. However, several issues need to be addressed that will improve the manuscript.

Comments:

1. It is hard to follow the study design from the main manuscript body- a separate study protocol is attached, however the inclusion of an overview figure, depicting key time points and individual sample numbers for each screening stage would be very helpful
2. The authors' description of their sample numbers is inconsistent and unclear. Each time, they describe the numbers in the study, the number is different (and usually lower) than the previous time it is provided.
In general, all the different study numbers for the separate screening stages need be explained and justified in the main body of the manuscript- Specifically, the separately attached study design initially describes about 1360 participants, yet in the results, it says that 1006 participants were enrolled (I93). Later, the immunogenicity response was determined for 964 (I135) or 760 (I138), and 470 samples (I147) were used to screen against VoCs. The authors must discuss why they only used have of their samples, what happened with the rest?
3. In the 'Safety Outcomes', the authors need to state the individual outcomes that were investigated. Section I118-129 is missing the % information
4. In the 'immunogenicity' section, the authors need to state and justify the individual sample numbers used for each assay; currently, it is hard to understand why different sample numbers were used for the individual assays. It is also unclear to what overall sample numbers the individual percentages (%) are referring. Another example is 'n=62' of participants in I201, is this number referring to n=1006 or n964.
5. Result section I160- I168 is very hard to follow- this needs clarification and maybe an additional

figure?

6. The Methods and Materials section needs to be improved, it needs to be more precise and complete. For example, it is not possible from the current description for anyone to repeat the neutralization assay that was used in the study- which system was used, etc.? This is especially a problem, because the authors claim that pseudotyped-neutralization assays are 'more sensitive' than live virus neutralization assays (l196-l200).

7. Often references are missing throughout the manuscript, e.g. l241 'Consistent with previous studies'- no reference; discussion about the VoCs; Material and Methods section are also missing references.

8. Fig. 3 should be consistent with Fig.1 and Fig.2 and show individual data points

9. Fig.4 needs error bars, how was the statistical analysis done without error bars?

10. Figure legends need to be more descriptive and informative; the sample number should be stated, statistical test should be described.

Point-by-point response to Reviewers

Reviewer #1 (Remarks to the Author):

This manuscript attempts to characterize cross-reactive responses to SARS-CoV-2 emerging variants after vaccination with an inactivated virus vaccine.

1. The manuscript is severely under referenced and several sentences contain claims without any citations. The authors should consider citing primary literature more.

Response: The authors greatly appreciate the reviewer's reminder. We have updated and cited more related published studies in the **Introduction** and **Discussion** section.

2. In figure 1, it becomes clear that antibody responses mature and increase after 2nd dose. However, the authors should include negative controls (naive individuals), convalescent individuals (previously infected) as well as individuals who received a different vaccine. This will really be the ultimate test of the assay setup as well as tell us how well the immune responses compare to other vaccines.

Response: The authors thank the reviewer for this suggestion, which can largely improve our manuscript. To ensure the accuracy of the specific antibody assay against SARS-CoV-2, we included a panel of 571 naïve individuals with neither prior COVID-19 symptoms nor a history of SARS-CoV-2 vaccination as negative controls. They were 18-59 years of age, and tested negative for SARS-CoV-2 viral nucleic acid. The overall description and the specific antibody assay results of this cohort were added in the **Methods** (page 13, line 351-354) and **Results** section (page 6, line 150-152).

We also included a panel of 16 convalescent sera to further confirm the reliability of the assay. Convalescent sera taken at sixth-month post symptom onset were selected by matching age from a cohort of 22 COVID-19 recovered patients in our published study¹. All individuals were 18-59 years of age, previously tested positive for the SARS-CoV-2 viral nucleic acid and diagnosed as COVID-19, and without a history of SARS-CoV-

2 vaccination at the sampling time. It turned out that they had a median SARS-CoV-2 specific antibody level of 246.80 (89.09-366.10), which was significantly higher than that of vaccine recipients after the second dose. The information about this group of subjects was added in the **Methods** (page 13, line 358-365) and **Results** section (page 8, line 203-208).

As the reviewer pointed out, it would be better if a parallel study could be conducted. However, most of the vaccine recipients in our country have been inoculated with the inactivated vaccine, while other types of vaccine, such as mRNA vaccine, are rarely available. Therefore, the major contribution of our work is to reveal the neutralizing activity induced by the inactivated vaccine BBIBP-CorV, especially against predominant variants of concern.

3. It is very surprising that the neutralization activity was significantly decreased against the B.1.1.7 pseudovirus. This is contrary to several other reports which use samples from mRNA vaccine recipients. To strengthen the claim, the authors should include 10 or more convalescent individuals in these experiments.

Response: We agree with the reviewer. Although our result is contrary to several other reports using samples from mRNA vaccine recipients, it is noteworthy that the reduced neutralizing activity against Alpha (B.1.1.7) is consistent with a previous finding, which revealed a marked decrease in GMTs in neutralization of variant Alpha in 25 inactivated vaccine recipients². The possible explanation for the inconsistency may be the different mechanism of vaccines. Briefly, mRNA vaccines encode a transmembrane-anchored SARS-CoV-2 full-length spike protein, stabilized in the prefusion conformation^{3,4}, they can amplify and sustainably express spike antigens to efficiently activate the immune system. Whereas inactivated virus vaccines retain almost all the antigens and epitopes of the virus⁵, but inactivation results in lowered immunogenicity and tendency to stimulate a weaker immune response than live viruses. These vaccines are incapable of replication, and normally require multiple immunizations to establish long-lasting immunity⁶.

To strengthen this claim, we also involved 16 convalescent sera to test the neutralizing activity against the wild-type strain and predominant variants. The results revealed that all convalescent sera neutralized variant Alpha (B.1.1.7) with a 1.9-fold reduction compared with the wild-type strain, which was comparable to the reduction in vaccine-elicited sera (2.2-fold reduction) in this study. The neutralization of convalescent sera against multiple variants was shown in Figure 3c and the **Results** section (page 8, line 209-216).

4. How do the authors feel about the drop of neutralization against B.1.351 but not P.1? They both are very similar in terms of mutations. Again, positive controls are necessary to strengthen these claims.

Response: Thanks for the reviewer's comments. Indeed, variant Beta (B.1.351) and variant Gamma (P.1) are very similar and share common amino acid mutations, including D614G, E484K, N501Y, and L18F. However, both of them contain other defining mutations. For example, Beta VOC contains mutations such as K417N, A701V, D80A, and D215G, while Gamma VOC carries mutations such as K417T, P26S, and H655Y. The possible explanation for the greater drop of neutralization against variant Beta is that N-terminal domain (NTD) substitutions in Beta variant contribute to neutralization escape⁷, and other undefined mutations of variant Gamma may compensate for the negative effects on inhibition of the E484K mutations according to the previous research⁸. Other studies have the same finding that variant Beta are more refractory to neutralization than variant Gamma^{9,10}. The discussion about this issue was added in the revised manuscript (page 11-12, line 306-310).

Similarly, in the convalescent sera panel, lower neutralizing antibody titres were observed against variant Beta (4.1-fold reduction) than variant Gamma (2.2-fold reduction), which was consistent with the results found in vaccine-elicited sera in this study.

Reviewer #2 (Remarks to the Author):

In this manuscript, Yu et al. assessed the safety and immunogenicity of an inactivated SARS-CoV-2 vaccine BBIBP-CorV, and measured the vaccine's efficacy against four global variants of concern: Lineage B.1.1.7, Lineage B.1.351, Lineage P.1, and Lineage B.1.526 to determine the neutralizing activity of vaccine-elicited sera. This is a very important study since the BBIBP-CorV vaccine has been approved and is being used for vaccinations. However, several issues need to be addressed that will improve the manuscript.

Comments:

1. It is hard to follow the study design from the main manuscript body- a separate study protocol is attached, however the inclusion of an overview figure, depicting key time points and individual sample numbers for each screening stage would be very helpful.

Response: The authors thank the reviewer for this suggestion. Indeed, it would be helpful to include an overview profile of this study, which was added as Figure 1. In this flowchart, we depicted the key time points and the sample size at each time point.

2. The authors' description of their sample numbers is inconsistent and unclear. Each time, they describe the numbers in the study, the number is different (and usually lower) than the previous time it is provided.

In general, all the different study numbers for the separate screening stages need be explained and justified in the main body of the manuscript- Specifically, the separately attached study design initially describes about 1360 participants, yet in the results, it says that 1006 participants were enrolled (193). Later, the immunogenicity response was determined for 964 (1135) or 760 (1138), and 470 samples (1147) were used to screen against VoCs. The authors must discuss why they only used have of their samples, what happened with the rest?

Response: The authors totally agree with the reviewer's suggestion. Briefly, the estimated sample size is 1360, however, depending on the vaccine supply and the actual vaccination rate, we finally enrolled 1006 eligible participants who were willing to receive two doses of inactivated SARS-CoV-2 vaccine BBIBP-CorV, and the immunoassays were performed in vaccine recipients who had blood sample taken. As shown in the newly added flowchart (Figure 1), 964 participants tested specific antibodies against SARS-CoV-2 on day 21 after the first dose, and 760 participants had specific antibody immunoassay and neutralization assay on day 28 after the second dose. Among the 760 vaccine recipients, 470 participants who had positive neutralizing activity against the wild-type strain and agreed to test neutralizing activity against multiple variants were included in neutralization assay against predominant variants. We have described in more detail in the revised manuscript for a better understanding.

3. In the 'Safety Outcomes', the authors need to state the individual outcomes that were investigated. Section I118-129 is missing the % information.

Response: We apologize for the missing information. No serious adverse events have been reported to date. All adverse reactions noted in this study were mild or moderate in severity and most of them resolved by day 7 after vaccination (page 5, line 113-114). The % information has been added (page 6, line 133-146).

4. In the 'immunogenicity' section, the authors need to state and justify the individual sample numbers used for each assay; currently, it is hard to understand why different sample numbers were used for the individual assays. It is also unclear to what overall sample numbers the individual percentages (%) are referring. Another example is 'n=62' of participants in I201, is this number referring to n=1006 or n964.

Response: We apologize for the unclear statement. The 62 participants refer to those who did not successfully induce neutralizing antibodies against SARS-CoV-2 among the 760 participants who had neutralization assay at day 28 after the second dose. To

avoid misunderstanding, we changed the statement to be consistent with the description in the result section as “The neutralizing antibody responses against SARS-CoV-2 were successfully induced in 698 (91.84%) of 760 individuals on day 28 after the second dose”, and listed the overall sample size after each percentage (%) in the context. A flowchart was also added (Figure 1) as suggested by the reviewer to depict the key time points and the sample size at each point.

5. Result section 1160- 1168 is very hard to follow- this needs clarification and maybe an additional figure?

Response: The authors thank the reviewer for this suggestion. We have added a Venn diagram as Figure 4 to show the cross reactivity of neutralizing antibody against four variants, and this part was simplified and re-described in the revised manuscript for better understanding (page 7-8, line 183-190).

6. The Methods and Materials section needs to be improved, it needs to be more precise and complete. For example, it is not possible from the current description for anyone to repeat the neutralization assay that was used in the study- which system was used, etc.? This is especially a problem, because the authors claim that pseudotyped-neutralization assays are ‘more sensitive’ than live virus neutralization assays (1196-1200).

Response: The authors thank the reviewer for the comment. Indeed, there is no exact evidence that the pseudovirus neutralization assays are more sensitive than the live-virus neutralization assays. Previous study revealed a high degree of concordance between pseudotype neutralization assay and isolated live SARS-CoV-2 neutralization assay¹¹, so we deleted the claim that pseudotype neutralization assays are ‘more sensitive’ from the **Discussion** section, and changed the statement to “pseudotype neutralization assay could also be used for evaluating the inhibitory effect of vaccine on viral attachment and entry”. Meanwhile, we improved the **Methods** section,

especially the “pseudovirus based neutralization assay” part, to make it more precise and complete for easy repeating the assay (page 14-15, line 379-397).

7. Often references are missing throughout the manuscript, e.g. 1241 ‘Consistent with previous studies’- no reference; discussion about the VoCs; Material and Methods section are also missing references.

Response: We apologize so much for these missing points. We have updated and cited more primary literature in the revised manuscript.

8. Fig. 3 should be consistent with Fig.1 and Fig.2 and show individual data points.

Response: The authors thank the reviewer for the comment. We have changed the type of the original Figure 3 (renumbered as Figure 5) to a scatter dot plot with error bars, which can display the individual data points and data distribution. It is noteworthy that we added a flowchart as Figure 1 and a Venn diagram as Figure 4, and renumbered the other figures (the original Figure 1 as Figure 2, the original Figure 2 as Figure 3).

9. Fig.4 needs error bars, how was the statistical analysis done without error bars?

Response: We apologize for the incorrect displaying format. The original Figure 4 (renumbered as Figure 6) has also been changed to be consistent with the original Figure 1 (renumbered as Figure 2) and the original Figure 2 (renumbered as Figure 3) in the revised manuscript as a scatter dot plot with error bars, and the error bars indicate the median with interquartile range (IQR).

10. Figure legends need to be more descriptive and informative; the sample number should be stated, statistical test should be described.

Response: Thanks so much for the comment. We reorganized the figure legends to be more descriptive by adding necessary information, including sample size and statistical analysis used.

Reference:

1. Liu, C., *et al.* Characterization of antibody responses to SARS-CoV-2 in convalescent COVID-19 patients. *J Med Virol* **93**, 2227-2233 (2021).
2. Wang, G.L., *et al.* Susceptibility of Circulating SARS-CoV-2 Variants to Neutralization. *N Engl J Med* **384**, 2354-2356 (2021).
3. Walsh, E.E., *et al.* Safety and Immunogenicity of Two RNA-Based Covid-19 Vaccine Candidates. *N Engl J Med* **383**, 2439-2450 (2020).
4. Jackson, L.A., *et al.* An mRNA Vaccine against SARS-CoV-2 – Preliminary Report. *N Engl J Med* **383**, 1920-1931 (2020).
5. Xia, S., *et al.* Safety and immunogenicity of an inactivated SARS-CoV-2 vaccine, BBIBP-CorV: a randomised, double-blind, placebo-controlled, phase 1/2 trial. *Lancet Infect Dis* **21**, 39-51 (2021).
6. Shin, M.D., *et al.* COVID-19 vaccine development and a potential nanomaterial path forward. *Nat Nanotechnol* **15**, 646-655 (2020).
7. Wibmer, C.K., *et al.* SARS-CoV-2 501Y.V2 escapes neutralization by South African COVID-19 donor plasma. *Nat Med* **27**, 622-625 (2021).
8. Chen, R.E., *et al.* Resistance of SARS-CoV-2 variants to neutralization by monoclonal and serum-derived polyclonal antibodies. *Nat Med* **27**, 717-726 (2021).
9. Lustig, Y., *et al.* Neutralising capacity against Delta (B.1.617.2) and other variants of concern following Comirnaty (BNT162b2, BioNTech/Pfizer) vaccination in health care workers, Israel. *Euro Surveill* **26**(2021).
10. Liu, Y., *et al.* Neutralizing Activity of BNT162b2-Elicited Serum. *N Engl J Med* **384**, 1466-1468 (2021).
11. Case, J.B., *et al.* Neutralizing Antibody and Soluble ACE2 Inhibition of a Replication-Competent VSV-SARS-CoV-2 and a Clinical Isolate of SARS-CoV-2. *Cell Host Microbe* **28**, 475-485 e475 (2020).

Editorial Note: Reviewer #1 was unavailable to review this round, so Reviewer #2 was asked to address the previous concerns raised.

REVIEWER COMMENTS

Reviewer #2 (Remarks to the Author):

The authors satisfactorily addressed all the comments and critics.

I went through the critics from reviewer #1 and the authors' responses.

There were altogether 4 main critic points:

#1 The authors updated their references and added relevant literature

#2 The authors added datapoints for naïve individuals and datapoints with convalescent serum, as requested by the reviewer (the figure and figure legend, as well as result and method sections were updated accordingly)

#3 the data observed for B.1.1.7 neut assay was discussed and to strengthen the results additional data with convalescent serum was added in Fig.3C; the Result and Discussion section were updated accordingly

#4 the data observed for B.1.351 neut assay was discussed and to strengthen the results additional data with convalescent serum was added in Fig.3C; the Result and Discussion section were updated accordingly

The initial concerns have adequately addressed in the revised manuscript.

REVIEWERS' COMMENTS

Reviewer #2 (Remarks to the Author):

The authors satisfactorily addressed all the comments and critics.

Response: Thanks for the approval of our manuscript.

I went through the critics from reviewer #1 and the authors' responses.

There were altogether 4 main critic points:

#1 The authors updated their references and added relevant literature

#2 The authors added datapoints for naïve individuals and datapoints with convalescent serum, as requested by the reviewer (the figure and figure legend, as well as result and method sections were updated accordingly)

#3 the data observed for B.1.1.7 neut assay was discussed and to strengthen the results additional data with convalescent serum was added in Fig.3C; the Result and Discussion section were updated accordingly

#4 the data observed for B.1.351 neut assay was discussed and to strengthen the results additional data with convalescent serum was added in Fig.3C; the Result and Discussion section were updated accordingly

The initial concerns have adequately addressed in the revised manuscript.

Response: The authors thank the reviewer for the comprehensive summary of the previous raised concerns and the affirmation to our manuscript.